# Mechanisms of *Caspases 3/7/8/9* in the Degeneration of External Gills of Chinese Giant Salamanders (*Andrias davidianus*)

**DOI:** 10.3390/genes13081360

**Published:** 2022-07-29

**Authors:** Shijun Yang, Caixia Tan, Xuerong Sun, Xiong Tang, Xiao Huang, Fan Yan, Guangxiang Zhu, Qin Wang

**Affiliations:** College of Life Science, Sichuan Agricultural University, Ya’an 625014, China; ysjsicau@hotmail.com (S.Y.); tcx19960507@163.com (C.T.); sunsun123@163.com (X.S.); 18583409571@163.com (X.T.); hx949987268@163.com (X.H.); yf2843933878@163.com (F.Y.); zhugx0711@163.com (G.Z.)

**Keywords:** metamorphosis, *caspase* gene, external gill, apoptosis, phylogeny

## Abstract

Metamorphosis is a critical stage in the adaptive development of amphibians from aquatic to terrestrial animals. Metamorphosis of the Chinese giant salamander is mainly manifested by the loss of external gills with consequent changes in the respiratory pattern. The loss of the external gill is regulated by the pathway of apoptosis in which caspase genes are the key factors. This study cloned and expressed the *caspase 3/7/8/9 genes* of the Chinese giant salamander. The main results were as follows: the complete open reading frames (ORFs) were 885 bp, 960 bp, 1461 bp and 1279 bp, respectively; *caspase 3/7/8/9 genes* all contained the CASc domain, and most of the motifs were located in CASc domain; and caspase 8 possessed two DED structural domains and caspase 9 possessed a CARD structural domain. Furthermore, results from the tissue distribution analysis indicated that *caspase 3/7/8/9 genes* were all significantly expressed in the external gill, and at 9 and 10 months of age (MOA), which is the peak time for the loss, the EXPRESSION level of *caspase 3/7/8/9 genes* was obviously high, which was consistent with the histological result. Moreover, the loss of external gills of the Chinese giant salamander may result from activation of both the apoptosis-related death receptor pathway and the mitochondrial pathway. Finally, it was discovered that thyroid hormone (TH) treatment could both advance the time point at which the external gills of the Chinese giant salamander began to degenerate and shorten this process. Interestingly, at the peak of its metamorphosis (9 MOA), the Chinese giant salamander further accelerated the metamorphosis rate of TH treatment, which suggested a promotive effect on the loss of external gills via the superimposition of the exogenous TH and *caspase* genes. The study of *caspase* genes in this experiment was conducive to understanding the mechanism of external gill loss in the Chinese giant salamander, as well as improving our understanding of the metamorphosis development of some Caudata species.

## 1. Introduction

The Chinese giant salamander is the largest amphibian and is endemic to China. It has been listed as an endangered species by the International Union for the Conservation of Nature and Natural Resources since the 1980s. The Chinese giant salamander has three pairs of dark pink external gills in the juvenile stage, which are the main breathing organs of the young salamanders [1]. The external gills exchange gases with the water to maintain an adequate oxygen level or adequate oxygenation. Other than respiratory organs, they are also involved in regulating the osmosis of water and ion homeostasis [2]. Metamorphosis in the Chinese giant salamander is characterized by progressive and complete regression of the external gills, with a consequent switch to respiration by the lungs [3]. The external gills of Chinese giant salamanders generally begin to decrease gradually from 9 to 16 months of age until complete regression, and their decreasing rate is related to the rate of dissolved oxygen, water temperature, breeding density and individual differences to some extent [1]. However, relevant studies on the molecular mechanism of the loss of the Chinese giant salamander’s external gills are still lacking.

Apoptosis, also known as programmed cell death (PCD), is a fundamental biological process that regulates the growth, development and immune response of multicellular organisms, and it is also a form of cell death that is highly regulated by genes [4,5]. Two alternative pathways activate apoptosis: activating the mitochondrial and activating the death receptor. The former triggers a form of apoptosis that is generally mediated by intracellular signals, which is also known as endogenous apoptosis [6]. By contrast, extrinsic pathway apoptosis is activated by several death receptors on the cell surface [6]. The activation and execution of apoptosis are regulated by the apoptotic (*caspase*) and B-cell lymphoma/leukemia 2 (*Bcl-2*) gene families [7,8]. The *caspase* family can directly cause the disintegration of apoptotic cells, and thus, plays a key role in the mechanism of apoptosis [9]. A total of fourteen kinds of *caspases* have been identified. They fall into two categories based on the gene location and function, namely, the inflammatory caspases (including *caspases 1*, *4*, *5*, *11*, *12*, *13* and *14*) and the apoptotic *caspases*. In addition, the apoptotic caspases can be further divided into initiators (including *caspases 2*, *8*, *9* and *10*), which are involved in the upstream cascade, and executioners (including *caspases*
*3*, *6* and *7*), which are located downstream of the cascade [10,11]. *Caspase 3* and *caspase 7* are two key effector *caspases* that play important roles in apoptosis-related pathways. The inactive *caspase 3* and *caspase 7* exist as inactive zymogens, and when activated by the initiators, they may inactivate many structural and functional proteins in the cell, leading to apoptosis [12]. *Caspase 8* is the initiator gene of the death receptor pathway and *caspase 9* is the initiator gene of the mitochondrial pathway; they are both located upstream of the two pathways and are mainly responsible for transmitting and amplifying apoptotic signals, thus activating apoptosis effector genes in the downstream of the pathway and inducing apoptosis [13]. 

The thyroid hormone (TH) signal is the cornerstone of molecular events that mediate profound morphological changes in early vertebrate development [14]. The TH consists mainly of prohormone thyroxine (T4) and 3,3′,5-triodothyronine (T3). The main substance synthesized and secreted by the thyroid gland is T4, while the main physiological role is played by T3 [15]. The TH at suitable concentrations improves the rates of embryo survival, hatching and juvenile survival in sturgeon [16,17]. Similarly, under suitable environmental conditions, the TH can stimulate amphibians to initiate the transition from larval to juvenile stages [15]. Metamorphic processes include the coordinated maturation and remodeling of organs; the neogenesis of limbs; the degeneration of tails; and the consequent changes in behavior, diet and ecological status [18]. Previous studies indicated a significant increase in T4 and T3 at the peak of metamorphosis in salamanders [19,20]. Moreover, exogenous TH stimulation on pre-metamorphic tadpoles can accelerate the whole process of metamorphosis and induce them to complete metamorphosis ahead of time [21,22]. Furthermore, the aquatic environment is increasingly contaminated by anthropogenic factors, such as pharmaceuticals, personal care products, and industrial and agricultural chemicals. These substances may directly or indirectly affect the TH, and thus, cause earlier or delayed metamorphosis in amphibians, as well as influence their health and survival [15,23]. Moreover, as the TH can even be directly detected in municipal wastewater [24], exploring the effects of TH on the metamorphosis of amphibians is also important in the conservation of populations affected by environmental pollution.

In this study, *caspases 3/7/8/9* in the Chinese giant salamander were cloned and characterized for the first time, and the relationships between the expression pattern of *caspases 3/7/8/9* and the external gill loss of the Chinese giant salamander were investigated. In addition, exogenous TH was used to induce metamorphosis in the Chinese giant salamander and the correlation between the expression pattern of *caspases 3/7/8/9* and external gill degeneration under hormone stimulation was investigated. This study not only contributed to furthering our understanding of the gene function and structures of *caspases 3/7/8/9* in Chinese giant salamanders, giving us a deeper insight into the molecular mechanism of external gill degeneration during their metamorphosis, but also provides important indications for studying the metamorphosis development in some Caudata species. Furthermore, as the Chinese giant salamander is considered a 350-million-year-old living fossil that links aquatic organisms to terrestrial ones, the study of the loss of its external gills is important for investigating the animals’ evolution “from aquatic to terrestrial” [25].

## 2. Materials and Methods

### 2.1. The TH Treatment Method and Sample Collection

The Chinese giant salamanders used in this study were all captive-bred individuals from the Yunxing Professional Giant Salamander Breeding Cooperative in Ya’an, Sichuan Province. As the Chinese giant salamander lays eggs only once a year, we selected 200 healthy juveniles that were 5 months of age (MOA) for rearing and follow-up studies to avoid sample attrition during the breeding process. After that, we randomly divided the 200 individuals into two groups: the natural group and the thyroid hormone treatment group, with 100 individuals in each group. 

In the natural group, we sampled the heart, liver, skin and muscle tissues of Chinese giant salamanders at a total of 14 time points from 7−20 MOA, and the external gills were sampled at 16 time points from 5−20 MOA, with each time point containing 3 biological replicates (*n* = 3). One side of the external gill tissues was stored at −80 °C for total RNA extraction. The other side of the external gill was fixed in 4% paraformaldehyde solution for 24–48 h to be used for a subsequent TUNEL assay.

In the TH treatment group, Chinese giant salamanders were exposed to 30 nM of T4 (Sangon Biotech Co., Ltd, Shanghai, China), which was based on physiological levels (10–50 nM) [15] and pre-experiments. We conducted a two-stage trial (Figure 1). Previous studies showed that Chinese giant salamanders generally begin to lose their external gills at 9 MOA [1]. Therefore, in the first stage, 8-MOA individuals were treated with the TH and their external gills were sampled at 30 (9 MOA), 60 (10 MOA) and 90 (11 MOA) days after treatment (DAT). Tissues were sampled from the area around the gill cavity when there were no visible external gills. To investigate whether the TH treatment would be affected by its natural metamorphosis, we advanced the time point to start the TH treatment by one month. In the second stage of the experiment, we started the TH treatment on 7-MOA individuals and sampled more intensively until 10 MOA. At this stage, we took samples every 5 days and obtained a total of 18 time points of sample data. The preservation and use of samples at both stages in this group were the same as that of the natural group. The Chinese giant salamanders not used in this study were kept for use in other experiments.

### 2.2. Cloning of Caspases 3/7/8/9

Total RNA was extracted from the external gills of Chinese giant salamanders by using RNAiso^TM^ Plus and reversely transcribed into cDNA by using PrimeScriptTMRT Reagent Kit with gDNAEraser (Takara Bio Inc, Japan) in accordance with an agreement with the manufacturer. Five pairs of primers (Table 1) were designed for the following study. PCR reactions were performed in a 30 μL reaction volume with the following cycling conditions: a 5 min initial denaturing at 95 °C, followed by 35 cycles of 30 s at 95 °C; 30 s annealing at *Tm* °C; then a *t* extension at 72 °C (*Tm* and *t* were adjusted according to the primers and sequence length of different genes), followed by a final 10 min extension at 72 °C. All the specific PCR products were purified and then inserted into a pClone007 Simple Vector and propagated in DH5α Competent Cells. Finally, 4−8 positive clones of each gene were sequenced by Sangon Biotech Co., Ltd.

### 2.3. Sequencing and Phylogenetic Analysis

The molecular weight and theoretical isoelectric point of *caspase 3/7/8/9 genes* were estimated by the use of the ProtParam tool (https://web.expasy.org/protparam/, accessed on 14 March 2022). The BLAST program of NCBI (http://www.ncbi.nlm.nih.gov/blast/, accessed on 14 March 2022) was used to search for sequences and perform homologous sequence alignment analysis to determine the open reading frames (ORFs) of the *caspase 3/7/8/9 genes* of the Chinese giant salamander. MAFFT v7.490 was used for sequence alignment in this study [26]. The amino acid sequence alignment results of *caspases 3/7/8/9* (sequences are listed in Appendix A) were generated through the use of the online software ESPript (https://espript.ibcp.fr/ESPript/ESPript/, accessed on 14 March 2022). MEME (http://meme-suite.org/, accessed on 14 March 2022) [27] was used to predict the conserved motif and the online software CDD (http://www.ncbi.nlm.nih.gov/cdd, accessed on 14 March 2022) was used to predict the conserved domain; then, TBtools was used for visualization [28]. SWISS-MODEL Repository software was used to predict the protein tertiary structure (https://swissmodel.expasy.org/, accessed on 14 March 2022) [29]. IQ-tree v2.1.3 was used to perform phylogenetic analysis based on the sequences that were aligned with the parameters: iqtree2-s input.phy-m MFP-B 1000--bnni-T AUTO (sequences are listed in Appendix A) [30]. Subsequently, iTOL (https://itol.embl.de/, accessed on 16 March 2022) [31] was used to visualize the phylogenetic tree.

### 2.4. Expression Analysis

Primers used for real-time quantitative PCR (RT-qPCR) were designed using Primer 5.0 and validated by using simple PCR amplification and electrophoresis (Table 1). The cDNAs of all Chinese giant salamanders obtained in Section 2.2 were used as templates; the target *caspase 3/7/8/9 genes* and the internal reference genes (β-actin and GAPDH) were amplified using fluorescence quantification, and three replicates of each gene were spiked per tissue. Using the 2^−ΔΔCt^ method, the qPCR data were analyzed to compare the expression differences of *caspase 3/7/8/9 genes* in the external gills of the Chinese giant salamander between the natural group and the TH treatment group at different time points. Finally, GraphPad Prism8 software was used to draw the graphs.

### 2.5. TUNEL Assay

Paraffin tissue blocks were dewaxed in water and then incubated with proteinase K for 30 min at 37 °C in an incubator, followed by the addition of an osmotic wash buffer for 20 min at room temperature. Next, the blocks were incubated with reaction solution (TdT/dUTP, 1:9) for an additional 2 h at 37 °C, followed by DAPI staining for 10 min protected from light. Tissue sections were then dried and sealed with neutral resin, and TUNEL-positive cells were observed under a microscope.

## 3. Results

### 3.1. Characterization and Sequencing: Analysis of caspases 3/7/8/9

The complete ORFs of *caspases 3/7/8/9* (GenBank accession no. OK078888/OK078889/OK078890/OK078891) were 885 bp, 960 bp, 1461 bp and 1279 bp, which encoded 294, 319, 486 and 422 amino acids, respectively. The molecular weights for these proteins were 33.03, 36.01, 55.49 and 46.98 kDa, respectively, and the predicted isoelectric points were 5.98, 5.11, 5.69 and 5.92, respectively. The molecular formulae of the *caspase 3/7/8/9 genes* were C_1450_H_2254_N_396_O_451_S_18_, C_1589_H_2459_N_425_O_487_S_22_, C_2448_H_3878_N_668_O_743_S_29_ and C_2068_H_3295_N_585_O_624_S_20_, respectively. Multiple sequence alignment showed high homology between the Chinese giant salamander *caspases 3/7/8/9* protein sequences and other species (Figure 2A,B). Moreover, the *caspases 3/7/8/9* proteins all had P20 large subunits and P10 small subunit structures, which were contained in the CASc domain. Caspase 8 contained two DED domains, each containing 80 amino acid sequences, and caspase 9 contains a CARD domain with 89 amino acid sequences (Figure 2A,B). A total of 10 motifs were predicted, most of which were located in the CASc structural domain. Two additional motifs existed in caspase 9 located in the CARD structural domain. The results of the protein tertiary structure showed that *caspases 3/7/8/9* in the Chinese giant salamander all consisted of α-helixes, β-sheets and random coils (Figure 2C).

Phylogenetic results showed that *caspases 3/7/8/9* clustered well into separate clades. Based on the analysis of the genes of distinct clades, the Chinese giant salamander is a sister to other Caudata species. Furthermore, analyses of *caspase 3* and *caspase 9* showed that the Caudata clade was nested with other anuran amphibians. However, a study on *caspase 8* indicated its closer relationship with turtles and avians, and an analysis of *caspase 7* indicated that this clade was more closely related to mammals and teleosts (Figure 3).

### 3.2. External Gill Loss and Expression Analysis of Caspases 3/7/8/9 in the Natural Group

We used qPCR to obtain the expression in various tissues, including external gill, heart, liver, skin and muscle, during the natural metamorphosis of 7−20 MOA Chinese giant salamanders. Results from the heat map of tissue distribution analysis showed that there were significantly higher expressions of *caspase 3/7/8/9 genes* in the external gills of the Chinese giant salamanders. *Caspases 7/8/9* were partially expressed in the skin tissues at 7, 18 and 20 MOA. In contrast, there was hardly any *caspase 3* expressed in skin tissues. None or very few genes were expressed in the heart, liver and muscle tissues (Figure 4A). 

Furthermore, the expression of *caspase 3/7/8/9 genes* was analyzed at 16 time points (from 5−20 MOA). The results indicated that *caspase 3/7/8/9 genes* were highly expressed at 9 and 10 MOA, the time at which the external gills of the Chinese giant salamander began to degenerate. In other MOA samples, *caspases 8/9* had relatively high expressions at 8 and 16 MOA, along with *caspase 7* at 13 MOA and *caspases 7/8* at 19 MOA. However, *caspase 3/7/8/9 genes* showed very low or even no expression at 5 and 6 MOA, as well as at 20 MOA when the gill cavity was completely closed (Figure 4B).

Samples from 12 out of the 16 time points were used for the TUNEL assay. Consistent with the expression results, samples at 9 and 10 MOA showed a larger number of apoptotic cells. However, contrary to the expression results, some apoptotic cells were observed in the 6-MOA samples, while they were hardly observed in the 5-, 7- and 8MOA samples. Interestingly, more apoptotic cells were observed at 12 and 14 MOA in which *caspase 3/7/8/9 genes* were hardly expressed. It was difficult to observe apoptotic cells in the final stage of external gill loss (Figure 4C).

### 3.3. Analysis of External Gill Loss and Expression of Caspases 3/7/8/9 in the TH Treatment Group

Results for three time points were obtained in the first stage of this study, including 30, 60 and 90 DAT. In samples from all time points, except 90 DAT of the *caspase 9* gene, there were significant or slight downward regulation in the treated group in contrast to the control group (Figure 5A). Moreover, the morphology at all three time points showed consistent results and there were even no obviously visible external gills in the TH treatment group at 30 DAT. However, individuals in the natural group obviously had external gills at all three time points (Figure 5B). Similarly, some apoptotic cells could be spotted sporadically at all three time points, but no clear difference was found between them (Figure 5C).

From the above results, we speculated that it would take a shorter time for Chinese giant salamanders to lose their external gills under the TH treatment. Meanwhile, to observe whether the natural degradation of the external gills had an effect on the results of the TH treatment group, we started the TH treatment a month earlier than the previous stage and sampled at more intensive time points. The results showed that there were high expressions of *caspase 3/7/8/9 genes* at 5−15 DAT; after that, *caspase 3/7* expressions decreased significantly until the end of the treatment. However, the expressions of *caspases 8/9* reached the peak again at 60 and 65 DAT and then decreased (Figure 6A). The morphological photographs showed that at 25 DAT, only sporadic gill filaments could be seen in individuals in the TH treatment group, while at 30 days, there were no gill filaments. After 40 DAT, branchial arches disappeared too (Figure 6B). Furthermore, apoptotic cells were seen at all time points, except 35 and 55 DAT (Figure 6C). 

## 4. Discussion

### 4.1. Characterization and Phylogeny of Caspases 3/7/8/9

It is known that the two tandem DED domains in vertebrate *caspase 8* are capable of activating inactive zymogens [32]. As the same two contiguous DED domains also exist in the Chinese giant salamander *caspase 8* (Figure 2A,B), this may suggest that the auto-activation mechanism of *caspase 8* zymogens in the Chinese giant salamander is similar to that in other vertebrates. Similarly, the CARD domain of procaspase 9 and Apaf-1 can interact with each other, which is a key factor in the recruitment of *caspase 9* zymogen into the apoptotic complex and the subsequent activation of *caspase 9*. Therefore, the CARD domain plays an important role in the mitochondria-mediated endogenous apoptotic pathway [33,34]. As the Chinese giant salamander *caspase 9* also had a CARD domain consisting of 90 amino acid residues (Figure 2A,B), this may suggest that this structural domain of *caspase 9* in the Chinese giant salamander plays an important role in the mitochondrial pathway of metamorphosis in giant salamanders. Therefore, the functional speculation of the structural domains further illustrated that *caspases 8/9* in the Chinese giant salamander were initiators. Since both the histidine and the active cysteine sequences of the Caspase family exist in the P20 large subunit [35] and it is the same with these two factors of *caspases 3/7/8/9* in the Chinese giant salamander, it is obvious that the P20 large subunit is an essential region for the protein activity of its *caspases 3/7/8/9*. Moreover, although the P10 small subunit does not contain a conserved pentapeptide sequence or active site, both subunits are essential and play an important role in the catalytic process [36].

The phylogenetic results suggested that *caspases 3/7/8/9* of different species cluster into one clade (Figure 3), indicating that all four genes were in a highly conserved state. As *caspase 3* and *caspase 7* genes are effector caspases with similar structures and functions, there is a high homology between them [37]. The phylogenetic results also showed that the *caspase 3* clade is a sister to the *caspase 7* clade, which confirmed that *caspase 3* and *caspase 7* were highly homologous. The Chinese giant salamander was clustered with other Caudata species in all four genes to form the Caudata clade. Recent results still support the idea that Caudata and Anura are monophyletic [38]. However, the Caudata clade was more closely related to turtles and avians based on *caspase 8*, while it was more closely related to mammals and teleosts based on *caspase 7.* This may be related to the specific functions of these two genes or the different evolutionary rates between species, but we still need further studies to explain this.

### 4.2. Natural Metamorphosis and Caspases 3/7/8/9 Expressions

Previous studies on the caspase genes in amphibian metamorphosis have mainly focused on the genus *Xenopus*. Former studies indicated that in *X. laevis*, *caspase* genes increased in regressing the organs, tail and intestines during the climax of metamorphosis, where programmed cell death occurred [39]. Moreover, the expression of the *caspase 3* gene significantly increased in tail muscle cells [40]. All the four genes in our study were highly expressed in the external gills of the Chinese giant salamander during metamorphosis (Figure 4A), which suggested that *caspase 3/7/8/9 genes* were mainly related to the loss of external gills in the Chinese giant salamander during metamorphosis. In addition, *caspases 7/8/9* were highly expressed in the skin at 7, 18 and 20 MOA. Judging by this, we speculated that this may be related to the higher metabolic frequency of the skin and the auxiliary respiratory function of the skin in replacing external gills during the metamorphosis of the Chinese giant salamander.

As key genes in the apoptotic pathway, the *caspase* protease family plays an extremely important role in animal metamorphosis development. The expression of apoptosis-related genes in anuran amphibians reaches its peak in the tail muscle at the peak of metamorphosis. In *X. tropicalis*, *caspase* genes are strongly expressed in their tails and the *caspase 3* gene is activated through the upregulation of some cytoplasmic proteases, which promotes apoptosis of caudal myocytes. Furthermore, as the increased expression of *caspase 3* and *caspase 7* was associated with the increase of apoptotic cells in the tails [41], the *caspase 9* gene was expressed in the tail before metamorphosis and its expression increased during the peak of metamorphosis. Moreover, *caspase 8* was also significantly upregulated at the peak of metamorphosis [42]. All these indicated that both the mitochondrial pathway and the death receptor pathway are involved in eliminating the tail during metamorphosis in *X. tropicalis* [42]. In this study, the expression of *caspase 3/7/8/9 genes* in the Chinese giant salamander peaked in the external gills at 9 and 10 MOA (Figure 4B). This was consistent with the time point when the Chinese giant salamander begins to lose its external gills [1]. This indicates that the relative expressions of *caspases 3/7/8/9* reach their highest levels at the time point when the external gills of the Chinese giant salamander start to degenerate (9 and 10 MOA). A TUNEL assay also confirmed and revealed the existence of a large number of apoptotic cells at 9 and 10 MOA (Figure 4C). However, many TUNEL-positive cells were still present at time points that were not at the peak of apoptosis (e.g., 6, 14 and 17 MOA), which we speculated may have been due to cells undergoing anastasis [43]. In addition, the expression of *caspase 7* was temporarily elevated in 19 MOA of the Chinese giant salamander. As *caspase 7* can regulate a series of reactions leading to cell proliferation and tissue regeneration [44,45], we hypothesized that cell proliferation and other phenomena existed at the final stage of gill loss of Chinese giant salamanders, thus promoting the closure of the gill cavity and the completion of metamorphosis.

*Caspases 8/9* are key proteins that initiate the death receptor and mitochondrial pathways. They are able to shear and activate *caspases 3/7*, which are the executive proteins downstream of these pathways and cause apoptosis. The results of this study showed that the expressions of *caspases 3/7/8/9* shared the same pattern, that is, rising first, peaking at 9/10 MOA and then declining. Therefore, based on this study, it was hypothesized that *caspases 8/9* as initiators could activate the downstream executioners, i.e., *caspases 3/7*, to complete the loss of external gills. In the death receptor pathway, the death-inducing signaling complex (*DISC*) is a key factor in activating *caspase 8* and initiating the pathway cascade, which is formed by the binding of *FADD* proteins to the death receptor/ligand complex that mediates apoptotic signaling. Moreover, the DED domain at the N-terminal of FADD protein can bind to caspase 8; therefore, *FADD* plays a key role in the activation of *caspase 8* [46]. In the mitochondrial pathway, the initiation of apoptosis leads to mitochondrial dysfunction and changes in the permeability of the outer membrane, which leads to the release of *CytC* from mitochondria to activate caspase 9 [47]. This study examined the mRNA expression of these two upstream genes. The results showed that the expression trends of *FADD* and *Cytc* were consistent with those of *caspases 8/9* (Appendix A). Therefore, we speculated that *FADD* and *CytC* genes could activate the *caspase 8/9* genes in the Chinese giant salamander. In summary, we suggest that metamorphosis in the Chinese giant salamander is caused by the activation of *caspase 8* and *caspase 9* by *FADD* and *CytC*, respectively, which consequently initiates the death receptor pathway and the mitochondrial pathway, respectively, and this is consistent with the apoptotic pathway that causes tail degeneration during metamorphosis in *Xenopus* species [42,48].

### 4.3. Acceleration of Metamorphosis through TH Treatment 

The metamorphosis climax of anuran amphibian is marked by the resorption of the thyroid-hormone-dependent tadpole tail [42]. *X. tropicalis* tadpoles with knocked-out T3 receptor genes are unable to complete metamorphosis [49]. Moreover, the TH can initiate metamorphosis and regulate morphological changes, such as the resorption of gills and the caudal fin (or the whole tail), and skeletal remodeling [50,51,52]. The TH can improve the embryonic survival, hatching rate and juvenile survival of sturgeon [16,17], as well as cause early or delayed deformation of amphibians and have an impact on their health and survival [15,23]. Therefore, we explored the effects of TH treatment on the metamorphosis of the Chinese giant salamander. 

In the first stage, we initially treated 8 MOA Chinese giant salamanders with the TH. There was already no obvious visible external gill in the 30-DAT samples, and compared with the control group, the expression of *caspases 3/7/8/9* significantly decreased in the TH treated group at 30 DAT (Figure 5A,B). As mentioned above, the expression of *caspase* genes associated with apoptosis should have significantly increased at the peak period of metamorphosis. However, the opposite result occurred here. Therefore, we speculate that the TH treatment group misses the peak metamorphosis at 30 DAT. In addition, as we wanted to know whether the metamorphosis rate of the TH-treated Chinese giant salamander samples would accelerate when they were at the peak of metamorphosis, we conducted the second phase of the experiment. At this stage, the TH treatment was advanced by one month and more intensive sampling was conducted.

At this second stage, we performed TH treatment on 7-MOA salamanders. Among them, *caspases 3/7/8* all showed significantly higher expression at 5−10 DAT. The expression of *caspase 9* reached a peak at 15 DAT, while that of *caspase 8* peaked at 10 DAT. This may have been because the death receptor pathway was initiated earlier than the mitochondrial pathway when facing exogenous TH stimulation (Figure 6A). However, the expression patterns of their upstream genes (e.g., FADD and Cytc) are currently lacking and we need to fully identify these patterns in subsequent studies. The expressions of *caspases 3/7* dropped to their minima at 25 DAT and stabilized later. Interestingly, in the morphological photographs, we saw that at 25 DAT, the last gill filaments still existed, but not until 40 DAT do the branchial arches disappear completely (Figure 6B). Therefore, we speculate that *caspases 3/7* are more closely related to the loss of gill filaments, and the disappearance of branchial arches may be relevant to other genes (e.g., *Bid* and *Bcl-2*) related to a cascade reaction. Furthermore, as in the first stage, external gills were still significantly expressed at 30 DAT in the second stage. Therefore, we believe that at its metamorphosis peak (9 MOA), the TH treatment on the Chinese giant salamander further accelerated the metamorphosis rate. This suggested a promotive effect on the loss of external gills in the giant salamander through the superimposition of exogenous TH and its caspase genes 

In this study, the *caspase 3/7/8/9 genes* of the Chinese giant salamander were obtained by cloning for the first time. The molecular mechanism of the metamorphosis of the Chinese giant salamander was tentatively explored and a link between apoptosis caused by *caspase* genes and the loss of external gill was established. In addition, we investigated the effect of TH treatment on the metamorphosis of the Chinese giant salamander, which is an important indicator for its subsequent production and reproduction, as well as a possible factor of environmental stimulation. The exploration of the relationship between external gill disappearance and apoptosis in Chinese giant salamanders not only helps to discuss the role of caspase genes in the process of the disappearance of external gills in the Chinese giant salamander but also provides important indications for the study of metamorphosis development in some Caudata species. In subsequent studies, we will focus on apoptosis-related pathways to provide a more comprehensive understanding of the molecular mechanism of metamorphosis in the Chinese giant salamander. In addition, we will further improve our histological experiments (e.g., adding a TUNEL-positive control and excluding interference from cells undergoing anastasis) to better validate our results.

## 5. Conclusions

This study cloned and characterized the *caspase 3/7/8/9 genes* of the external gills of Chinese giant salamanders for the first time. A close relationship between *caspase 3/7/8/9 genes* and external gill loss was discovered. *Caspase 3/7/8/9 genes* were all significantly expressed in the external gills, and at 9 and 10 MOA, the peak time of the loss, the expression levels of the *caspase 3/7/8/9 genes* were obviously high. It was discovered that TH treatment could both advance the time point at which the external gills of the Chinese giant salamander began to degenerate and shorten this process. Moreover, at the peak of its metamorphosis (9 MOA), the Chinese giant salamander further accelerated the metamorphosis rate of TH treatment, which suggested a promotive effect on the loss of external gill via the superimposition of exogenous TH and its caspase genes. 

## Figures and Tables

**Figure 1 genes-13-01360-f001:**
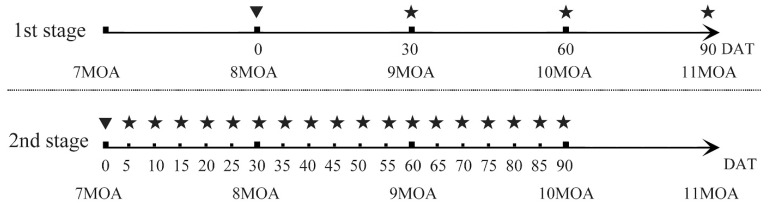
Sampling time points in the TH treatment group. A triangle represents the time point at which the TH treatment began; a pentagram represents a time point at which samples were taken.

**Figure 2 genes-13-01360-f002:**
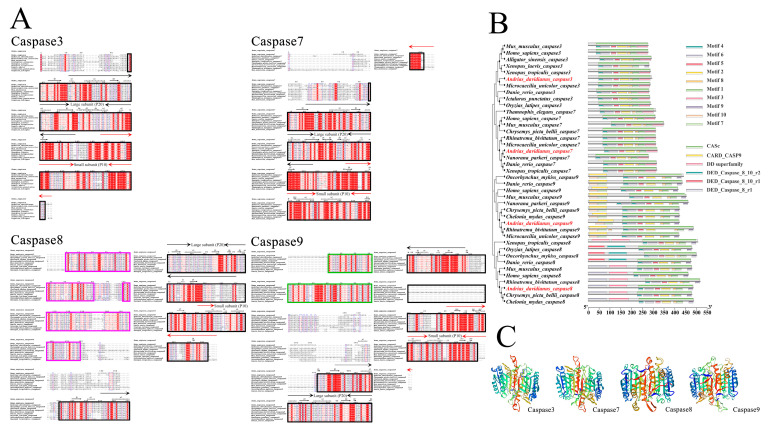
(**A**) Multiple sequence alignments of *caspases 3/7/8/9*. A black box indicates a CASc domain; a purple box indicates a DED domain; a green box indicates a CARD domain; a black arrow indicates a P20 large subunit; a red arrow indicates a P10 small subunit. (**B**) Structural domain and motif prediction. (**C**) The tertiary structures of caspases 3/7/8/9 in the Chinese giant salamander.

**Figure 3 genes-13-01360-f003:**
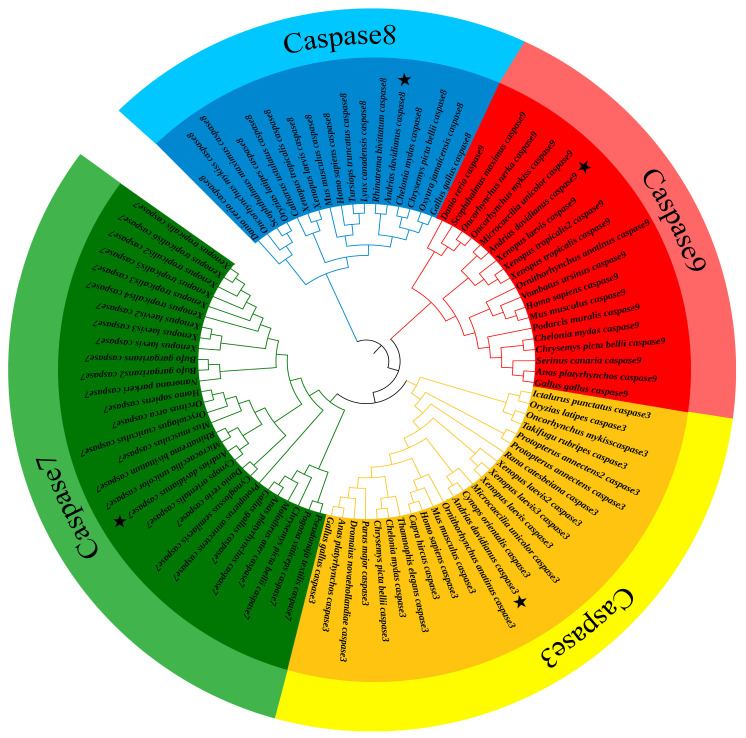
Phylogenetic tree of *caspases 3/7/8/9*. Pentagrams represent the Chinese giant salamanders used in this study.

**Figure 4 genes-13-01360-f004:**
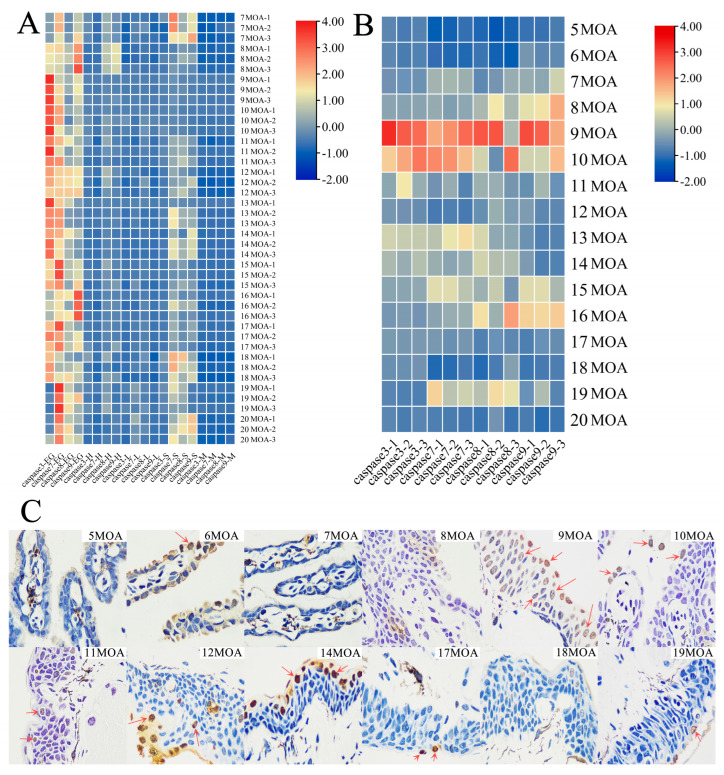
(**A**) The expression patterns of tissue distribution in the Chinese giant salamander *caspase 3/7/8/9 genes*. EG: external gill; H: heart; L: liver; S: skin; M: muscle. (**B**) The expression patterns of *caspase 3/7/8/9 genes* in the external gill of the Chinese giant salamander at different MOAs. (**C**) Apoptosis in the external gill of a Chinese giant salamander during metamorphosis, as seen using a TUNEL assay (400×). Red arrows indicate TUNEL-positive cells.

**Figure 5 genes-13-01360-f005:**
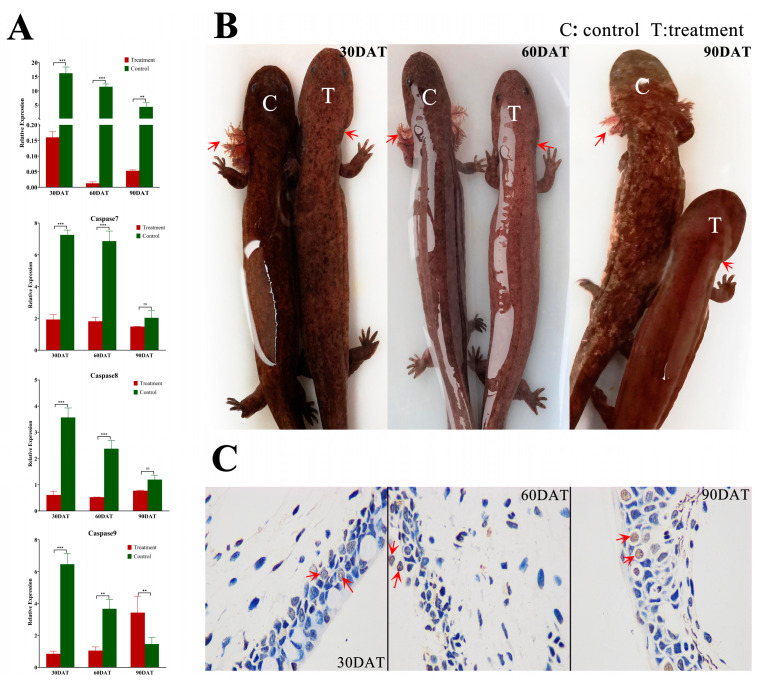
TH treatment results for 8-MOA Chinese giant salamanders at three sampling time points (30 DAT, 60 DAT and 90 DAT). (**A**) Relative expression levels of *caspases 3/7/8/9* in the external gills of the Chinese giant salamanders after the TH treatment. Asterisks indicate significant differences from the control groups: **, *p* < 0.01; ***, *p* < 0.001; ns, no significance. (**B**) Comparison of external gill morphology between the TH-treated and control groups. (**C**) Apoptosis in the external gill of the Chinese giant salamander after the TH treatment, as seen using a TUNEL assay (400×). Red arrows indicate TUNEL-positive cells.

**Figure 6 genes-13-01360-f006:**
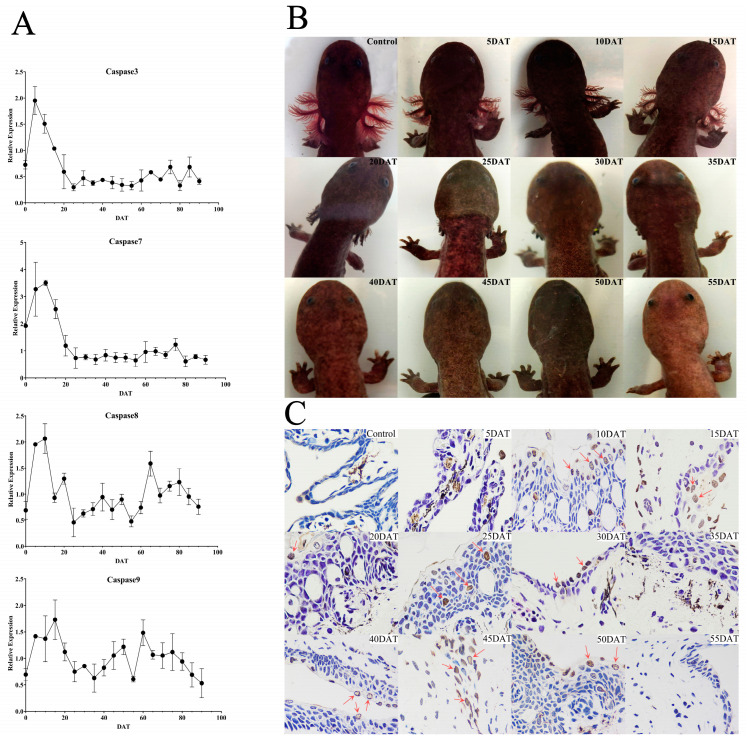
Results after the TH treatment for 7-MOA Chinese giant salamanders sampled every 5 DAT. (**A**) The expression trends of *caspases 3/7/8/9* in the external gills of the Chinese giant salamanders after the TH treatment. (**B**) Trends in external gill morphology of the Chinese giant salamanders after the TH treatment. (**C**) Apoptosis in the external gills of the Chinese giant salamanders after the TH treatment, as seen using a TUNEL assay (400×). Red arrows indicate TUNEL-positive cells.

**Table 1 genes-13-01360-t001:** Primer sequences were used in this study.

Primer Name	Primer Sequence	Primer Purpose
Caspase 3-F	5′-GAGGCAGCGAGGACTATTGT-3′	Caspase 3 amplification
Caspase 3-R	5′-TGGTGGCTCATTGTTCTTGTT-3′	
Caspase 7-F	5′-TTTTACCCGCCACCTCCTATCC-3′	Caspase 7 amplification
Caspase 7-R	5′-ACAACAGTAACACAGTTCCCCC-3′	
Caspase 8-F	5′-GATGACAAACCCCATGTAAGG-3′	Caspase 8 amplification
Caspase 8-R	5′-TCTCCCAAATGAAGGTGCTC-3′	
Caspase 9-F	5′-CTCATGTCCGGTACGGTAGA-3′	Caspase 9 amplification
Caspase 9-R	5′-CAGAGGTTTGTGACCGTATGC-3′	
M13-F	5′-CGCCAGGGTTTTCCCAGTCACGAC-3′	Universal primer
M13-R	5′-CAGCGGATAACAATTTCACACAGG-3′	
β-actin-F	5′-GCCGTGACCTGACAGACTACCT-3′	RT-qPCR
β-actin-R	5′-AGTCCAGGGCGACATAGCAGAG-3′	
GAPDH-F	5′-GACCACTGTCCACGCAGTCAC-3′	
GAPDH-R	5′-GATGTTCTGGTTGGCACCTCT-3′	
Q Caspase 3-F	5′-GGACATTGAGGCAAAGCCAGAA-3′	
Q Caspase 3-R	5′-TGAGGTTTCCAGCATCCACATC-3′	
Q Caspase 7-F	5′-GCAGATCCTCACCAGGGTCAAC-3′	
Q Caspase 7-R	5′-CGTCAGCATGGACACCACACAA-3′	
Q Caspase 8-F	5′-CAGACGGCAGATGTCCAACG-3′	
Q Caspase 8-R	5′-TATCATCACCTCTCGGGCAGC-3′	
Q Caspase 9-F	5′-TGGGCACCACTGTCCAACTC-3′	
Q Caspase 9-R	5′-ATCTCCGCTGTCCATTACCGA-3′	

## Data Availability

The data were deposited in GenBank with accession number OK078888/OK078889/OK078890/OK078891.

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
