# Peer review of "Mechanisms of Caspases 3/7/8/9 in the Degeneration of External Gills of Chinese Giant Salamanders (Andrias davidianus)"

_genes, 2022, doi:10.3390/genes13081360_

Round 1

Reviewer 1 Report

Thank you for submitting this interesting paper. 

Below some comments

Where you will find the term Suggestion, means I think some rewording can improve how the message is transferred to reader, but are not mandatory actions. It's just for your consideration.  

Lines 34-35: Chinese giant salamander has three pairs of dark pink external gills in the juvenile stage, which are the main breathing organs of the young salamanders. 

Please include reference. 

Lines 35-36: The external gills exchange gases with the water to meet giant sala- mander’s basic need for oxygen

Suggestion: to matin adequate oxygen level, or adequate oxygenation. 

Lines 38-40: The most im- portant features in the metamorphic development of the Chinese giant salamander are the decrease till loss of its external gills and its gradual change from gill respiration to lung respiration (Eo et al., 2012)

Suggestion: Metamorphosis in the Chinese giant salamander is characterised by progressive and complete regression of the external gills, with consequente switch of the respiration to the lungs. 

Line 42: months of age until totally disappear., and their

You have to choose, either . or , . Also, you already say that the gills are disappearing, no need to repeat. As terminology, in general in an animal nothing disappear, you can use either complete regression or complete atrophy, but disappearing no. 

Lines 47-49: Apoptosis, also known as programmed cell death (PCD), is a fundamental biological process that regulates the growth, development and immune response of multicellular organisms.

Reference, please

Line 49: It is also a form of death highly regulated by genes

I would say this is the main function. Never the less this sentence needs to be rephrased: It is a highly generally regulated for or CELL death. Please, you don't have to stick to my sentence, but use the word cells. 

Lines 50-51: Apoptosis is usually fulfilled through two pathways: activating the mitochon- drial or activating the death receptor.. 

These are two alternative pathways of apoptosis, so can be activated through pathways. Please also remove the double .. at the end of the sentence. 

Line 53: By contrast, exogenous apoptosis

Please replace exogenous with extrinsic pathway.

Lines 67-68: Caspase 3 precursor is located in the cytoplasm but plays an important role in nuclear changes in apoptotic cells.

I would say this is not essential. Never the less, if you want to write it than you need to be more specific and explaining that the inactive form is in the cytoplasm and the active forme is translocated to the nucleus and what dose there. 

Lines 69-70: Caspase 7 plays an even more important role in the loss of cell viability (Kamada et al., 2005; Walsh et al., 2008).

As it is, I don't understand the relevance of this sentence. How/Why is more important? 

Lines 85-87: Previous studies have indicated a significant increase in T4 and T3 at the peak of metamorphosis in sala- manders (Caudata) (Larras-Regard et al., 1981; Alberch et al., 1986).

LEt's start to say that Caudata is a clade and you should say it. But Caudata, dose't contains salamanders only. So either you make the sentence more generic or you switch talking about the order Urodela. 

Lines 90-92: Besides, the aquatic environment is increasingly contaminated by anthro- pogenic factors such as pharmaceuticals, care products, and industrial and agricultural chemicals.

Please consider to add personal to care products. 

Lines 119-120: In the natural group, we sampled the heart, liver, skin and muscle of Chinese giant salamanders respectively at a total of 14 time points from 7-20 MOA

Why not kidneys and lungs? I would have collected all the tissues, to be honest since you are sacrificing 200 animals. 

Lines 123-124: The other side of the external gill was fixed in 4% paraformaldehyde solution to be used for subsequent TUNEL assay.

For how long these tissues have been fixed?

Line 166: the online software

the online software CDD

Line 167: was used to predict conserved domain,,

Remove the double comma, please.

Line 184: 2.5. TUNEL assay

I have two questions here:

1) Do you have any evidence or do you have evaluated the coss-reactivity of the TUNEL test in the salamander? I see the results of the homology but I don't find any specific reference to the TUNEL 

2) Did you consider that also cells undergoing anastasis can be TUNEL positive, and the possibile impact on your interpretation? 

Line 217: The Caudata clade nested with other anuran amphibians. .

Please remove the double .

The histology part can be better described (e.g., location from which the pictures are taken), and I wander if a pathologist took a look at the TUNEL results. 

Reviewer 2 Report

The paper is generally well presented, but the English needs some further editing. I started by marking corrections on the attached file but did not have time to mark them all. The main issue is that when 'Chinese salamander' is written it usually should be 'the Chinese salamander'. In the plural, 'Chinese salamanders' is ok without the article, i.e. by itself.

The technical detail is beyond my level of knowledge but it makes sense to read.

please define all abbreviations, and separately in the abstract.

I think there is a lack of overview, placing the findings in the bigger picture of metamorphosis.  For example, the discussion could have a paragraph covering: what is the normal role of TH in metamorphosis (in other amphibians), does the level of TH increase at metamorphosis, is TH the next upstream driver in normal circumstances? Or is tissue sensitivity to TH increased at the time of metamorphosis, or both? Are any other activator of caspases known in other amphibians? The authors have shown nicely that the caspases are part of the effector mechanism, but the reader would like to know what are the upstream drivers, as far as can be hypothesised, and where the new findings fit into the puzzle of metamorphosis.

Please check the references and place generic/species names in italic font if the original publication has done that (all recent ones will have)
